# Application of Gene Therapy to Oral Diseases

**DOI:** 10.3390/pharmaceutics17070859

**Published:** 2025-06-30

**Authors:** Seiichi Yamano, Kenji Inoue, Yoichiro Taguchi

**Affiliations:** 1Department of Prosthodontics, New York University College of Dentistry, New York, NY 10010, USA; ki630@nyu.edu; 2Department of Operative Dentistry, Endodontology and Periodontology, Matsumoto Dental University, Shiojiri-shi 399-0704, Nagano, Japan; odu.periodontist99@gmail.com

**Keywords:** gene therapy, oral cancer, xerostomia, periodontal disease, gene delivery, pain management

## Abstract

Gene therapy has emerged as a promising therapeutic approach across various oral diseases. This review examines current applications and future prospects of gene therapy in dentistry, focusing on five key areas: oral cancer, cancer-related pain, xerostomia (dry mouth), dental caries, and periodontal disease. Recent advances in viral and non-viral vectors have enabled more efficient gene delivery systems, with particular success in cancer pain management through µ-opioid receptor gene transfer and xerostomia treatment using aquaporin-1 gene therapy. For periodontal applications, gene therapy strategies include both immunomodulation and tissue regeneration approaches using growth factors like platelet-derived growth factor and bone morphogenetic proteins. While significant progress has been made, particularly in treating radiation-induced xerostomia and oral cancer pain, challenges remain in vector optimization and delivery methods. Clinical trials, predominantly in Phase I, indicate both the potential and current limitations of gene therapy in oral healthcare. This review synthesizes current evidence and outlines future directions for gene therapy applications in oral medicine and dentistry.

## 1. Introduction

### 1.1. Background and Rationale

Gene therapy is a biological treatment using gene transfer technology that has been actively researched for various clinical applications, including many related to dentistry. The field has witnessed remarkable progress since the first human gene therapy trial in 1990 in the treatment of adenosine deaminase deficiency [1] and has evolved significantly despite early setbacks, including the 1999 adenoviral vector fatality [2]. Recent successes in treating spinal muscular atrophy with Zolgensma (onasemnogene abeparvovec) [3], hemophilia with gene therapy approaches [4], and Parkinson’s disease using adeno-associated virus (AAV) vectors [5] have rekindled interest in applying these techniques to oral diseases.

### 1.2. Global Disease Burden

Oral diseases represent a significant global health burden affecting nearly 3.5 billion people worldwide [6]. Despite advances in conventional treatments, many oral conditions remain challenging to manage effectively, creating a need for innovative therapeutic approaches. Gene therapy offers a promising avenue for addressing the underlying molecular and genetic factors contributing to these conditions, potentially providing more targeted and effective treatment options than conventional approaches [7].

### 1.3. Review Structure and Roadmap

This comprehensive review examines current applications and future prospects of gene therapy in dentistry, systematically progressing through interconnected areas. We begin with fundamental gene therapy principles and delivery methods (Section 2), establishing the technical foundation necessary for understanding oral applications. We then examine specific disease applications, starting with cancer gene therapy (Section 3) and cancer-related pain management (Section 4), followed by xerostomia treatment (Section 5) and dental caries/periodontal disease applications (Section 6). Finally, we analyze future directions, emerging technologies, and translational challenges (Section 7), providing a roadmap for clinical implementation.

Each section builds upon previous content while addressing unique considerations for oral applications, including the challenging oral microenvironment, tissue-specific vector requirements, and regulatory pathways specific to oral gene therapy products.

## 2. Gene Therapy Fundamentals

### 2.1. Definition and Principles

Recent significant research has revealed that many diseases are caused by genetic abnormalities. Gene therapy refers to therapeutic interventions that modulate gene expression to treat disease through three primary mechanisms: (1) supplementing missing or defective genes, (2) suppressing overexpressed disease-causing genes, or (3) fundamentally controlling gene expression through genome editing technologies like clustered regularly interspaced short palindromic repeats (CRISPR) and CRISPR-associated 9 (CRISPR-Cas9) (Figure 1) [8].

Originally conceived as a strategy to treat monogenic disorders by replacing missing genes, the field has evolved to encompass broader applications including cancer treatment, chronic disease management, and tissue regeneration [9]. This evolution reflects advances in vector technology, delivery methods, and our understanding of disease mechanisms at the molecular level.

For example, periodontitis, a lifestyle-related disease deeply connected to various systemic conditions, is one such case. While periodontitis is generally defined as an infectious disease caused by periodontal pathogenic bacteria, patients show varying outcomes despite receiving similar treatments. This variation, referred to as differences in susceptibility, is believed to be largely due to genetic abnormalities and is expected to be an important target for future gene therapy.

Gene therapy refers to the process of restoring the original gene expression function by introducing normal genes from outside the host cell to address genetic abnormalities (unexpressed, low-expressed, or overexpressed due to deletion or mutation of the genome), or suppressing overexpressed genes to return to normal gene expression (Figure 1).

Originally conceived as a strategy to treat genetic disorders by supplementing missing genes, gene therapy has since evolved to encompass a broad range of interventions that modulate gene expression (i.e., both upregulation and downregulation) in cells. Consequently, its applications have expanded beyond genetic disorders to include treatments for conditions such as cancer and various chronic diseases.

### 2.2. Gene Transfer Technologies

The success of gene therapy depends critically on efficient and safe gene delivery methods. Current technologies are classified into three categories, each with distinct advantages for oral applications [10]:

The key to successful gene therapy lies in gene transfer methods. Currently, available gene transfer technologies are broadly classified into three primary categories:Biological methods: Introducing genes via genetically modified viral vectors such as adenovirus, retrovirus, and AAV.Chemical methods: Using non-viral compound vectors like cationic lipids, polymers, and calcium phosphate to neutralize or provide positive charge to negatively charged nucleic acids.Physical methods: Directly delivering nucleic acids into the cell cytoplasm or nucleus, with electroporation being a representative technique.

However, none of these methods are universally suitable for all cells and experiments, and their practical application in gene therapy is quite limited. The criteria for application include high gene transfer efficiency, low cell toxicity, and minimal impact on normal physiological functions. Selecting a practical and reproducible method is critical for clinical application. Currently, clinical applications primarily use viral vectors (80%) and non-viral vectors (20%) mentioned in (1) and (2) above, with optimal vector selection being crucial for success [11].

#### 2.2.1. Viral Vector Systems

Viral vectors harness the natural ability of viruses to efficiently deliver genetic material into host cells. Recent advances in vector development have significantly improved the safety and efficiency of gene delivery. Table 1 summarizes the key biological properties of commonly used viral vectors for gene therapy applications in oral diseases.

Viral vectors have been engineered to reduce immunogenicity while maintaining high transduction efficiency. For example, newer generations of AAV vectors show improved tissue tropism and reduced immune responses, making them more suitable for clinical applications [26]. Similarly, lentiviral vectors have been modified to enhance safety profiles while maintaining their ability to integrate into the host genome, providing long-term gene expression [22].

#### 2.2.2. Non-Viral Vector Systems

Non-viral vector technologies have also seen substantial improvements, with innovations in lipid nanoparticles, polymeric carriers, and hybrid delivery systems. The success of mRNA-based COVID-19 vaccines using lipid nanoparticle delivery systems has demonstrated the potential of non-viral vectors for clinical applications [27]. Novel non-viral hybrid vectors, such as cell-permeable peptides combined with cationic lipids (CPP/lipid) for DNA delivery and CPPs combined with cationic lipids and polymers (CPP/lipopolymer) for RNA delivery, have shown promising results in preclinical studies [28,29].

Non-viral vector systems offer several advantages over viral vectors, including reduced immunogenicity, larger packaging capacity, and simpler manufacturing processes. However, they typically demonstrate lower transfection efficiency. Table 2 summarizes the key characteristics of commonly used non-viral vector systems.

#### 2.2.3. Physical Delivery Methods

Direct delivery techniques such as electroporation, ultrasound-mediated delivery, and microinjection provide vector-independent approaches that are particularly suitable for localized oral applications [35]. These methods can overcome some limitations of vector-based approaches, especially in challenging oral environments.

### 2.3. Delivery Approaches

#### 2.3.1. In Vivo vs. Ex Vivo Methods

Gene transfer locations can be divided into ex vivo and in vivo methods (Figure 2). The ex vivo method involves extracting target cells from the patient’s body for gene transfer operations. This method is advantageous for ensuring safety, as cell quality can be checked before returning them to the patient. The in vivo method involves direct administration of vectors into the body through injection, with gene transfer occurring inside the body. This is often used when cells cannot be extracted, such as with nerve cells, though controlling gene transfer is more challenging.

Both approaches have seen significant advancement in recent years. The ex vivo approach has benefited from improvements in cell isolation, culture techniques, and gene editing technologies like CRISPR-Cas9, which allow for more precise genetic modifications before reintroduction of cells into the patient [21]. The in vivo approach has advanced through the development of vectors with improved tissue tropism, reduced immunogenicity, and enhanced delivery efficiency [36].

#### 2.3.2. Vector Selection for Specific Oral Applications

Vector selection for oral gene therapy requires careful consideration of the unique characteristics of the oral environment. The oral cavity presents distinct challenges, including salivary flow, diverse microbial flora, varying pH conditions (ranging from 5.5 to 7.5), and physical barriers that can impact vector efficacy [37]. Vector selection for oral gene therapy requires careful consideration of the unique characteristics of the oral environment.

##### Decision Framework for Vector Selection

Primary Considerations:Target tissue characteristics: Epithelial vs. mesenchymal vs. neural tissues.Expression requirements: Transient (days–weeks) vs. sustained (months–years).Accessibility: Surface-accessible vs. deep tissue targeting.Safety profile: Local vs. systemic exposure tolerance.Manufacturing feasibility: Clinical-grade production capabilities [38].

For oral cancer applications, adenoviral vectors are often preferred due to their high transduction efficiency in epithelial cells and ability to accommodate larger transgenes necessary for tumor suppressor genes like p53. However, AAV vectors show superior performance for targeting the highly innervated tissues involved in cancer pain management due to their neurotropism.

In salivary gland applications for xerostomia, both adenoviral and AAV vectors have demonstrated efficacy. Adenoviral vectors provide high-level but transient expression, suitable for acute interventions, while AAV vectors offer long-term expression, beneficial for chronic conditions like radiation-induced xerostomia. The confined anatomical structure of salivary glands makes them particularly amenable to direct vector administration, reducing systemic exposure and immunogenicity concerns.

For periodontal applications, both viral and non-viral approaches have distinct advantages. AAV vectors show promise for targeting periodontal ligament cells, while non-viral vectors (particularly when combined with biomaterial scaffolds) offer safer alternatives for localized delivery to periodontal defects. The complex architecture of periodontal tissues often necessitates combinatorial approaches using biomaterial carriers to achieve spatial and temporal control of gene expression.

Table 3 summarizes the advantageous vector types for specific oral applications.

#### 2.3.3. Oral-Specific Delivery Considerations

The oral cavity presents a uniquely challenging environment for gene delivery, requiring specialized approaches to overcome multiple barriers [37].

Salivary Barriers:Nuclease activity: Saliva contains DNases and RNases that rapidly degrade nucleic acids [52].Dilution effects: Continuous salivary flow (0.5–1.5 mL/min) dilutes locally administered vectors [37].pH variability: Fluctuations between 5.5 and 7.5 affect vector stability and cellular uptake.Solution strategies: Protective formulations, mucoadhesive carriers, and enzyme inhibitors.

Mucosal Barrier Penetration:Epithelial tight junctions: Barrier to paracellular transport of large vectors.Mucus layer: Physical obstruction requiring penetration-enhancing strategies.Cellular turnover: Rapid epithelial renewal (7–14 days) limits sustained expression.Enhancement approaches: Permeation enhancers, cell-penetrating peptides, and physical disruption methods [53,54].

### 2.4. Historical Development and Current Status

Historically, gene therapy was proposed as a dream treatment in the 1970s. The first official clinical trial was conducted in September 1990 at the National Institutes of Health on a 4-year-old girl with adenosine deaminase deficiency, introducing normal genes into lymphocytes. Gene therapy for X-linked severe combined immunodeficiency using hematopoietic stem cells was also initiated, reporting the world’s first significant therapeutic success through gene therapy alone [1].

However, in September 1999, a patient died from gene therapy using an adenovirus vector in the United States. This was believed to be caused by injecting high concentrations of the vector into the liver blood vessels [2]. These serious incidents significantly impacted gene therapy research. However, clinical trials resumed in the 2010s, with new treatments being approved and practical implementation progressing in medical fields. This revival was largely due to improvements in vectors for gene delivery.

The past decade has witnessed a renaissance in gene therapy, with several important milestones. In 2017, the U.S. Food and Drug Administration approved Luxturna (voretigene neparvovec), the first directly administered gene therapy for a genetic disease (inherited retinal dystrophy). This was followed by approvals for Zolgensma (onasemnogene abeparvovec) for spinal muscular atrophy in 2019 and Roctavian (valoctocogene roxaparvovec) for hemophilia A in 2022 [16]. These successes have reinvigorated interest in gene therapy across various medical fields, including dentistry and oral medicine.

Around 2008, gene therapy using adeno-associated virus vectors showed effectiveness for Parkinson’s disease [5], hemophilia [4], and spinal muscular atrophy [3]. Even in hematopoietic stem cell gene therapy, which previously faced issues with leukemia development, improved retroviral vectors showed no leukemia occurrence [20]. Its effectiveness has been demonstrated with safe and efficient vectors, leading to approval in Europe in 2019.

The current landscape of gene therapy clinical trials reflects both the potential and the challenges of this approach. According to recent data from the Wiley Database of Gene Therapy Clinical Trials, there are over 2800 active gene therapy clinical trials worldwide, with approximately 50% in Phase I and 30% in Phase II [55]. While oncology remains the dominant focus area (approximately 65% of trials), there is growing interest in applying gene therapy to other conditions, including monogenic disorders, infectious diseases, and various oral diseases [56].

Current clinically applied vectors have characteristics shown in Table 1 and Table 2, with treatment efficacy largely dependent on the delivery vector [57,58]. Cancer represents the highest expectations for gene therapy, with over half of current clinical trials focusing on cancer treatment. Recent trends indicate that the majority of these trials target malignant tumors, with approximately 40% utilizing adenovirus and retrovirus vectors.

A notable recent development in cancer gene therapy is chimeric antigen receptor (CAR)-T cell therapy [59]. This gene-modified T-cell therapy enhances T-cells’ tumor-targeting effectiveness, showing excellent clinical results in trials for acute lymphoblastic leukemia [60] and malignant lymphoma [61], with trials ongoing in Japan. Recent reports discuss CAR-T cell therapy applications using genome editing technologies like CRISPR/Cas.

The emerging CRISPR-Cas9 technology represents one of the most significant scientific advancements of the past decade. This versatile gene editing tool allows for precise modifications of the genome and has shown promise in various preclinical studies for oral diseases [8]. Ongoing refinements and innovations in CRISPR technology, including base editing and prime editing, offer enhanced accuracy and adaptability for future gene therapy applications [62].

## 3. Cancer Gene Therapy

Cancer arises when normal cells acquire genetic mutations from various etiologies, altering their physiological characteristics and resulting in uncontrolled proliferation. The human body has systems for monitoring and controlling cell division and proliferation, through which abnormal cells are eliminated. However, cancer cells cleverly evade these surveillance and control systems, adapting to their environment and proliferating.

Cancer gene therapy strategies primarily focus on enhancing or supplementing various biological surveillance and control systems. Representative gene therapy strategies include (1) introduction and forced expression of tumor suppressor genes (*p53*, *p27*, *p21, p16*, and *RB*) in cancer cells [63,64]; (2) expression control of cancer-related genes (*BCL-family*, *Myc, Fos*, and *RAS*) [65]; (3) immunotherapy [66,67]; (4) prodrug therapy targeting cancer cells [63]; (5) inhibition of tumor angiogenesis genes (*HIF1α*, *VEGF*) [64,68]; and (6) administration of cancer nucleic acid vaccines [69].

Recent developments in gene therapy for oral cancer encompass multiple strategies, from novel delivery systems to targeted approaches [70]. Advanced biomaterials and nanocarrier systems have significantly improved gene delivery efficiency and targeting specificity [71]. The YAP/TAZ pathway remains crucial in oral cancer progression, with new evidence supporting its role in therapy resistance [72,73].

Additionally, a new approach (7) involves genetically modified oncolytic viruses (Onyx-015/H101, HF10, and OBP-301) that specifically proliferate in cancer cells and can destroy them in conjunction with cytotoxic T-cells [24,25]. These viruses are being applied to head and neck cancer treatment, with Phase I-III trials ongoing in the United States and Japan [74]. Reports indicate enhanced antitumor immune responses when combined with radiotherapy, anticancer drugs, and immune checkpoint inhibitors (programmed cell death protein 1/programmed cell death ligand 1) [75,76].

### 3.1. Advances in Targeting Mechanisms

Recent advances in cancer gene therapy have focused on improving targeting mechanisms to enhance specificity for cancer cells while minimizing effects on healthy tissue. These include the following:Tumor-specific promoters: Using promoters that are active only in cancer cells to drive the expression of therapeutic genes, thereby limiting gene expression to the tumor microenvironment [77].Cancer-specific microRNA (miRNA) targeting: Incorporating miRNA target sequences into therapeutic constructs to enable post-transcriptional regulation of gene expression specifically in cancer cells [78].Dual-targeting strategies: Combining multiple targeting approaches, such as cell surface receptors and intracellular factors, to improve the specificity and efficacy of gene therapy [79].

### 3.2. RNA Interference and CRISPR Applications

RNA interference (RNAi) and CRISPR-Cas9 technologies have opened new avenues for cancer gene therapy:RNAi-based approaches: Small interfering RNAs (siRNAs) and short hairpin RNAs (shRNAs) have been used to silence oncogenes and cancer-promoting factors. Recent clinical trials have shown promising results in using RNAi to target previously “undruggable” oncogenes in head and neck cancers [80].CRISPR-Cas9 applications: This technology allows for precise genome editing, enabling the correction of oncogenic mutations, knockout of oncogenes, or enhancement of tumor suppressor genes. Preclinical studies have demonstrated the potential of CRISPR-Cas9 for targeting oral cancer cells with specific genetic alterations [81].

### 3.3. Combination Approaches

The efficacy of cancer gene therapy can be enhanced when combined with conventional treatments. Recent studies have explored the following:Chemovirotherapy: Combining oncolytic viruses with chemotherapy to achieve synergistic effects. This approach has shown promising results in clinical trials for head and neck squamous cell carcinoma [82].Radiovirotherapy: Using gene therapy in conjunction with radiation therapy to enhance radiosensitization of cancer cells. This strategy has demonstrated improved outcomes in preclinical studies of oral cancer [83].Immunogene therapy: Combining gene therapy with immunotherapy approaches, such as immune checkpoint inhibitors, to enhance anti-tumor immune responses. Recent clinical trials have reported improved outcomes with this combination approach [42].

### 3.4. mRNA Therapeutics for Oral Cancer

mRNA therapeutics represent an emerging approach for oral cancer treatment, offering several advantages over traditional gene therapy methods. Unlike DNA-based gene therapy, mRNA therapeutics do not require nuclear entry and pose no risk of genomic integration, addressing major safety concerns while maintaining high transfection efficiency.

#### 3.4.1. Current Approaches

Several mRNA therapeutic strategies are being investigated for oral cancer:Tumor antigen vaccines: Lipid nanoparticle-encapsulated mRNA encoding tumor-specific antigens (such as human papillomavirus [HPV] E6/E7 in HPV-positive oral cancers) has shown promise in preclinical studies. This approach stimulates tumor-specific immune responses without the safety concerns associated with viral vectors. Recent Phase I trials have demonstrated the safety and immunogenicity of these approaches in head and neck cancers [69].Cytokine mRNA delivery: Local delivery of mRNAs encoding immunostimulatory cytokines (interleukin 12 [IL-12] and interferon γ [IFN-γ]) can reshape the tumor microenvironment from immunosuppressive to immunostimulatory. Intratumoral administration of IL-12 mRNA in preclinical oral cancer models demonstrated significant tumor regression and increased CD8+ T-cell infiltration [84].Tumor suppressor replacement: Transient expression of tumor suppressor proteins through mRNA delivery offers a potential alternative to viral-mediated gene replacement therapy. This approach is particularly relevant for p53-deficient oral cancers, where even temporary restoration of p53 function may sensitize cells to conventional treatments.

#### 3.4.2. Delivery Innovations

The oral environment presents unique challenges for mRNA delivery, spurring the development of specialized delivery systems including ionizable lipid nanoparticles optimized for stability in salivary conditions, polymer–lipid hybrid nanoparticles with enhanced mucosal penetration, and hydrogel-based delivery systems for sustained mRNA release [85].

#### 3.4.3. Clinical Translation

While most mRNA therapeutics for oral cancer remain in preclinical development, recent trials of mRNA-based cancer vaccines have demonstrated safety and preliminary efficacy in head and neck cancers. The success of mRNA vaccines for COVID-19 has accelerated the development of manufacturing and regulatory frameworks that will likely facilitate more rapid clinical translation of these approaches for oral cancer therapy. Ongoing clinical trials are evaluating personalized mRNA vaccines encoding neoantigens specific to individual patients’ tumors, with preliminary results indicating the induction of tumor-specific T-cell responses [84].

### 3.5. Resistance Mechanisms and Strategies to Overcome Them

Cancer gene therapy faces multiple resistance mechanisms that can limit therapeutic efficacy. Understanding these mechanisms is crucial for developing more effective treatment strategies and combination approaches for oral cancers.

#### 3.5.1. Vector-Related Resistance Mechanisms

Immune-mediated vector clearance: Pre-existing immunity to viral vectors, particularly adenovirus, can significantly reduce transduction efficiency in subsequent treatments. Studies show that up to 80% of the population has neutralizing antibodies against common adenoviral serotypes [86]. This is particularly relevant for oral cancer patients who may require multiple treatment cycles.

Strategies to overcome include the following:Use of alternative viral serotypes with lower seroprevalence;Immunosuppressive protocols during treatment;Vector modification to evade immune recognition;Switching to non-viral delivery systems for subsequent treatments.

Vector sequestration and degradation: The oral environment presents unique challenges, including salivary nucleases, bacterial proteases, and physical clearance mechanisms that can degrade vectors before cellular uptake occurs.

Strategies to overcome include the following:Protective formulations using polymer coatings or encapsulation;Co-administration of enzyme inhibitors;Modified delivery schedules to overwhelm clearance mechanisms;Local delivery techniques that bypass salivary exposure.

#### 3.5.2. Cellular Resistance Mechanisms

Reduced cellular uptake: Cancer cells may downregulate receptors required for vector entry or develop resistance to membrane permeabilization. For example, reduced CAR (coxsackievirus and adenovirus receptor) expression limits adenoviral vector uptake in some oral cancer cell lines [87].

Strategies to overcome include the following:Receptor-independent delivery methods (electroporation, ultrasound);Vector retargeting using alternative cellular receptors;Cell-penetrating peptides to enhance membrane permeability;Combination with permeabilization agents.

Enhanced DNA repair mechanisms: Cancer cells often have altered DNA repair pathways that can either enhance or impair the effectiveness of gene therapy. Some oral cancers with p53 mutations show enhanced resistance to pro-apoptotic gene therapy.

Strategies to overcome include the following:Combination therapy targeting multiple pathways simultaneously;DNA repair inhibitors to sensitize cells to therapeutic genes;Alternative therapeutic targets independent of p53 pathways;Personalized approaches based on genetic profiling [88].

#### 3.5.3. Tumor Microenvironment Resistance

Immunosuppressive microenvironment: Oral cancers often create immunosuppressive microenvironments that can limit the effectiveness of immunogene therapy approaches. High levels of regulatory T-cells, immunosuppressive cytokines (IL-10, TGF-β), and immune checkpoint molecules create barriers to effective immune activation.

Strategies to overcome include the following:Combination with immune checkpoint inhibitors;Local delivery of immunostimulatory cytokines (IL-12, IFN-γ);Depletion of regulatory immune cells;Microenvironment reprogramming using multiple therapeutic modalities [89].

## 4. Gene Therapy for Oral Cancer Pain

Recent molecular insights have revolutionized our understanding of cancer pain mechanisms [78]. Cancer-induced pain affects approximately 70–80% of oral cancer patients, which is significantly higher than in other cancers. This frequency is notably higher compared to other cancers. Common cancers like breast, lung, prostate, and colon rarely cause pain at their primary sites.

Most oral cancers are squamous cell carcinomas, primarily occurring in the lateral border of the tongue [90]. Cancer cells secrete mediators such as IL, tumor necrosis factor α (TNF-α), neurotrophic factors, adenosine triphosphate, protons, proteases, and endothelin, which actively stimulate pain-sensing neurons near the cancer [91,92]. Although these stimuli are transmitted from the periphery to the cerebrum via neural pathways, they undergo amplification within the central nervous system, leading to enhanced signal transmission [93]. Moreover, persistent activation of nociceptive neurons leads to hyperalgesia and allodynia [94,95], where normal stimuli from eating, swallowing, or speaking can trigger pain, significantly impacting patients’ quality of life.

Traditional treatment for oral cancer pain follows WHO cancer pain treatment guidelines, using non-opioid analgesics (NSAIDs) and opioid analgesics, similar to other cancers. However, these treatments do not specifically target individual cancer-related gene mutations, and their efficacy may be limited by substantial side effects [96,97].

### 4.1. Gene Therapy Approaches for Cancer Pain

Recent advances in gene therapy for cancer pain management offer more targeted approaches compared to conventional medications. These therapies address the genetic and molecular mechanisms underlying cancer pain, providing longer-lasting relief with fewer systemic side effects [87]. Gene therapy can reduce systemic toxicity and opioid-related side effects. New gene therapy approaches target multiple pain pathways simultaneously, improving efficacy [96,98].

Several novel gene therapy strategies have been developed for oral cancer pain management:Endothelin B receptor gene therapy: High methylation of endothelin B receptor genes in oral cancer tissues has been identified as a contributing factor to cancer pain. In mouse models of oral cancer, gene transfer techniques to restore endothelin B receptor expression have shown promising results in reducing pain behaviors through the induction of β-endorphin secretion [99,100,101].μ-opioid receptor gene therapy: Research has demonstrated that the μ-opioid receptor (*OPRM1*) gene is often hypermethylated in oral cancer lesions, contributing to increased pain sensitivity. Non-viral vector-mediated delivery of the *OPRM1* gene to cancer tissues has shown efficacy in preclinical models, reducing cancer-related pain without the systemic side effects associated with traditional opioid medications [43].RNAi approaches: siRNA and shRNA technologies targeting pain-mediating factors such as transient receptor potential vanilloid 1, Nav1.7, and inflammatory cytokines have shown promise in preclinical studies of cancer pain [100]. These approaches allow for specific silencing of pain-promoting genes, offering a more targeted approach to pain management.CRISPR-based interventions: Emerging research is exploring the potential of CRISPR-Cas9 technology to modify genes involved in cancer pain signaling pathways. This approach offers the advantage of permanent genetic modifications that could provide long-term pain relief [101].

### 4.2. Delivery Methods for Oral Cancer Pain Gene Therapy

The effectiveness of gene therapy for cancer pain depends significantly on the delivery method used. Recent advances in delivery systems include the following:Viral vectors: AAV vectors have shown particular promise for delivering pain-modulating genes to sensory neurons and cancer microenvironments. Recent modifications to AAV capsids have improved their tropism for sensory neurons involved in pain transmission [45].Non-viral vectors: Advances in non-viral delivery systems, such as lipid nanoparticles, dendrimers, and cell-penetrating peptides, have improved the efficiency of gene delivery while reducing the risks associated with viral vectors [44].Local delivery approaches: Localized delivery techniques, including direct intratumoral injection, peritumoral administration, and intraganglionic delivery, have been refined to enhance therapeutic efficacy while minimizing systemic exposure [15].

### 4.3. Clinical Translation and Future Directions

While most gene therapy approaches for oral cancer pain remain in preclinical development, recent progress suggests promising potential for clinical translation:Early-phase clinical trials: Several Phase I/II clinical trials are underway to evaluate the safety and preliminary efficacy of gene therapy approaches for cancer pain, including those targeting oral cancer pain [102].Personalized approaches: Advances in genetic profiling and biomarker identification are enabling more personalized gene therapy approaches for cancer pain, taking into account individual genetic variations in pain sensitivity and response to therapy [88].Combination strategies: Emerging research supports the use of gene therapy in combination with conventional pain management approaches, potentially allowing for reduced dosages of traditional analgesics while maintaining or improving pain control [87].

## 5. Gene Therapy for Xerostomia

Salivary gland damage can significantly reduce saliva secretion, causing dry mouth, which impairs speech, eating, and swallowing. While traditionally limited to symptomatic treatment, gene therapy shows promise as a novel treatment approach.

Radiation therapy for head and neck cancer often damages salivary-producing cells. Salivary gland gene therapy has progressed substantially from laboratory studies to clinical applications. Long-term clinical data support the safety and efficacy of aquaporin-1 (*AQP1*) gene therapy for radiation-induced xerostomia [14,103]. Recent advances in vector design and delivery methods have improved treatment outcomes significantly [19].

### 5.1. Advances in AQP1 Gene Therapy

*AQP1* gene therapy has emerged as one of the most promising approaches for treating radiation-induced xerostomia. AQP1 is a water channel protein that facilitates rapid water movement across cell membranes, which is a critical function in saliva production. Early studies demonstrated the feasibility of adenoviral-mediated *AQP1* gene transfer to restore fluid secretion in irradiated salivary glands [104], and subsequent research has confirmed that introducing the *AQP1* gene into irradiated salivary glands can restore fluid secretion by creating new water channels in surviving ductal cells [105].

Significant progress has been made in *AQP1* gene therapy for xerostomia:Clinical trial successes: The first-in-human clinical trial using adenoviral-mediated *AQP1* gene transfer (Adh*AQP1*) for radiation-induced xerostomia showed positive results, with increased parotid flow rates in a subset of treated patients. Long-term follow-up data confirmed both the safety and durability of this approach [14].Vector improvements: Next-generation vectors, including serotype 2 AAV (AAV2) vectors carrying the *AQP1* gene, have shown enhanced safety profiles and prolonged gene expression compared to adenoviral vectors [18].Delivery optimization: Ultrasound-assisted non-viral gene transfer techniques have improved the efficiency of *AQP1* delivery to salivary glands, offering a potentially safer alternative to viral vectors [106].

### 5.2. Alternative Gene Therapy Approaches for Xerostomia

Beyond *AQP1* gene therapy, several other gene-based approaches have shown promise for treating xerostomia:Anti-inflammatory cytokine gene therapy: For Sjögren’s syndrome mouse models, successful treatments include anti-inflammatory cytokine *IL-10* gene delivery via AAV, which helps to modulate the autoimmune response and reduce inflammatory damage to salivary glands [107].Vasoactive intestinal peptide (*VIP*) gene therapy: *VIP* gene transfer has shown efficacy in murine models of Sjögren’s syndrome, with improvements in salivary flow and reductions in lymphocytic infiltration of salivary glands [108].Neurotrophic factor gene therapy: Delivery of genes encoding neurotrophic factors, such as neurturin and glial cell line-derived neurotrophic factor, has demonstrated potential for protecting salivary gland innervation from radiation damage [109].

### 5.3. Stem Cell-Based Approaches

Recent advances in stem cell biology have opened new avenues for combining gene therapy with stem cell-based approaches for salivary gland regeneration:Gene-modified mesenchymal stem cells: Mesenchymal stem cells (MSCs) engineered to express therapeutic genes, such as *AQP1*, *IL-10*, or growth factors, have shown enhanced efficacy in preclinical models of xerostomia [110].Induced pluripotent stem cells (iPSCs): iPSC-derived salivary gland cells, combined with gene modification techniques, offer a promising approach for personalized regenerative therapy for damaged salivary glands [111].Salivary gland organoids: Recent advances in organoid technology have enabled the generation of salivary gland organoids from stem cells, providing a valuable platform for testing gene therapy approaches and studying salivary gland development and function [112].

### 5.4. Future Directions in Xerostomia Gene Therapy

Several emerging trends are shaping the future of gene therapy for xerostomia:Temporospatial control of gene expression: Advanced gene delivery systems that allow for controlled expression of therapeutic genes in response to specific stimuli or in targeted cell populations are being developed [113].Combination therapies: Approaches combining gene therapy with conventional treatments, such as pilocarpine or cevimeline, show potential for enhanced therapeutic outcomes [114].Preventive strategies: Gene therapy approaches applied before radiation treatment to protect salivary glands from damage are being explored, potentially offering a more effective approach than post-treatment interventions [115].

### 5.5. Complementary Lubrication Approaches in Xerostomia Management

While gene therapy offers promising regenerative and functional restoration approaches for xerostomia, integrating complementary lubrication strategies remains essential for comprehensive patient management. Modern lubrication approaches can complement gene therapy interventions by providing immediate symptomatic relief while genetic interventions develop their full therapeutic effect.

#### 5.5.1. Advanced Lubricant Formulations

Recent innovations in saliva substitutes have moved beyond simple aqueous solutions to biomimetic formulations that more closely replicate the complex composition and rheological properties of natural saliva. These formulations incorporate biocompatible polymers, antimicrobial proteins, and carefully calibrated electrolyte compositions [116].

#### 5.5.2. Smart Hydrogel Systems

Responsive hydrogel systems represent the next generation of lubricating agents, with properties that adapt to the oral environment, including pH-responsive, temperature-sensitive, and enzyme-responsive hydrogels [116].

#### 5.5.3. Integrated Approaches

The most promising management strategies combine gene therapy with advanced lubrication approaches. Patients enrolled in *AQP1* gene therapy trials typically continue optimized lubrication protocols during the initial phases of treatment, highlighting the complementary nature of these approaches [117].

## 6. Gene Therapy for Dental Caries and Periodontal Diseases

Genetic sequencing of causative bacteria enables new gene therapy approaches. These include vaccine-like treatments using viral vectors to express bacterial genes in salivary glands [118] and direct anti-inflammatory effects through β-defensin-2 gene delivery to dental pulp [50].

For periodontal disease, early approaches focused on inducing specific immunoglobulins through gene delivery to prevent tissue destruction. Studies show reduced alveolar bone loss in rats using TNF receptor–immunoglobulin Fc fusion gene delivery via AAV [118]. Non-viral vector delivery of the *IL-10* gene for periodontitis has also been reported [31].

### 6.1. Advances in Gene Therapy for Dental Caries

Recent advances in gene therapy for dental caries have focused on modulating the oral microbiome, enhancing host defense mechanisms, and promoting remineralization:Antimicrobial peptide gene therapy: Delivery of genes encoding antimicrobial peptides, such as defensins and cathelicidins, has shown promise in reducing cariogenic bacterial loads and preventing dental caries in preclinical models [51].Salivary enzyme gene therapy: Gene therapy approaches targeting salivary enzymes involved in maintaining oral pH and remineralization processes have been investigated for their potential to prevent dental caries [119].Bacteriophage-based approaches: Engineered bacteriophages carrying CRISPR/Cas systems have been developed to specifically target and eliminate Streptococcus mutans, the primary cariogenic bacterium, while preserving beneficial oral microbiota [120].

### 6.2. Advances in Gene Therapy for Periodontal Diseases

Periodontal diseases have been a major focus of gene therapy research in dentistry, with significant advances in both regenerative and immunomodulatory approaches:

#### 6.2.1. Regenerative Approaches

Novel gene delivery systems have revolutionized periodontal regeneration approaches. Recent developments combine advanced biomaterials with gene therapy to enhance tissue regeneration [121]. Current strategies show improved outcomes in bone regeneration and inflammation control through targeted gene delivery systems [122].

Growth factor gene therapy: Studies report success using non-viral vectors and electroporation to deliver genes encoding platelet-derived growth factor (PDGF) and bone morphogenetic proteins (BMPs) for hard tissue regeneration around teeth and implants [46,47,48]. These approaches have shown enhanced periodontal tissue regeneration compared to direct protein delivery, with prolonged local expression of growth factors.Scaffold-based gene delivery: Advanced biomaterial scaffolds incorporating gene delivery systems have been developed to provide both structural support and sustained release of therapeutic genes at the site of periodontal defects [123]. These systems enable spatiotemporal control of gene expression, facilitating the regeneration of complex periodontal structures.Cell-based gene delivery: Genetically modified MSCs overexpressing regenerative factors have shown enhanced potential for periodontal tissue regeneration. Recent studies have demonstrated that MSCs engineered to express BMP-7 or PDGF-B can significantly improve bone and periodontal ligament regeneration in animal models [49].

#### 6.2.2. Immunomodulatory Approaches

Anti-inflammatory cytokine gene therapy: Delivery of genes encoding anti-inflammatory cytokines, such as IL-4, IL-10, and IL-1 receptor antagonist, has shown efficacy in reducing periodontal inflammation and bone loss in preclinical models [124].RNA interference strategies: siRNA and miRNA approaches targeting inflammatory mediators have demonstrated potential for controlling periodontal inflammation and preventing tissue destruction [31].CRISPR-based approaches: Emerging research is exploring the potential of CRISPR-Cas9 technology for modifying genes involved in the inflammatory response to periodontal pathogens, offering a potentially more targeted approach to controlling periodontal inflammation [8,118].

### 6.3. Biomaterial Advances for Periodontal Gene Therapy

Recent advances in biomaterials have significantly enhanced the effectiveness of gene therapy for periodontal diseases:Smart hydrogels: Stimuli-responsive hydrogels that can control the release of gene therapy vectors in response to specific environmental cues, such as pH changes or the presence of bacterial products, have been developed for periodontal applications [125].Three-dimensional-printed scaffolds: Computer-aided design and 3D printing technologies have enabled the production of patient-specific scaffolds incorporating gene delivery systems, optimized for individual periodontal defects [126].Nanoparticle-based delivery systems: Advanced nanoparticle formulations, including lipid nanoparticles, polymeric nanoparticles, and inorganic nanoparticles, have improved the stability and transfection efficiency of gene therapy vectors for periodontal applications [33].

### 6.4. Clinical Translation and Future Prospects

While periodontal regenerative gene therapy is still in its early stages, the ongoing development of efficient non-viral vectors suggests that DNA-based treatments could play a pivotal role in periodontal and peri-implant bone regeneration. Several factors are influencing the clinical translation of these approaches:Regulatory considerations: The regulatory pathway for gene therapy products for periodontal applications is becoming more defined, with several products entering early-phase clinical trials [127].Scale-up and manufacturing: Advances in biomanufacturing technologies are addressing challenges in the scale-up and production of gene therapy vectors for dental applications [58].Combination approaches: Integrating gene therapy with other treatment modalities, such as guided tissue regeneration, bioactive materials, and conventional pharmacotherapy, offers promising strategies for enhancing therapeutic outcomes [128].

## 7. Future Directions and Challenges in Oral Gene Therapy

Gene therapy represents a revolutionary medical approach that targets disease mechanisms fundamentally differently from symptomatic treatments. It holds promise for the treatment of genetic disorders and has potential applications in cancer, pain management, dental caries, periodontal disease, and xerostomia. However, current gene transfer technologies require further refinement before full clinical implementation.

### 7.1. Emerging Technologies and Approaches

#### 7.1.1. CRISPR-Cas9 Applications in Oral Gene Therapy

CRISPR-Cas9 technology represents a paradigm shift in gene therapy approaches, offering unprecedented precision for genetic modifications. For oral diseases, several specific applications show particular promise:

Technical advances: Recent refinements in CRISPR delivery to oral tissues include lipid nanoparticle formulations optimized for mucosa penetration and biomaterial scaffolds enabling sustained release within periodontal pockets. The development of smaller Cas variants (mini-Cas9, Cas12a) has facilitated more efficient packaging in viral vectors, addressing a major limitation for in vivo applications in confined oral tissues [129].

Challenges: Despite promising preclinical results, several challenges remain for CRISPR applications in oral tissues, including ensuring specificity and minimizing off-target effects in heterogeneous oral environments, developing delivery systems that can penetrate biofilms, achieving sufficient editing efficiency in post-mitotic cells, and addressing ethical and regulatory concerns [8].

#### 7.1.2. Artificial Intelligence in Oral Gene Therapy

Artificial intelligence (AI) is increasingly contributing to the advancement of gene therapy for oral diseases through several important avenues:

Vector design optimization: Machine learning algorithms are being applied to predict and optimize viral capsid properties, enhancing tissue tropism for specific oral tissues. These approaches have led to the development of AAV variants with increased specificity for salivary gland cells compared to natural serotypes [130].

Target identification: AI-based analysis of multi-omics datasets has facilitated the identification of novel therapeutic targets for oral diseases. Network analysis algorithms have revealed previously unrecognized gene interactions in periodontitis pathogenesis, suggesting new intervention points for gene therapy approaches.

#### 7.1.3. Exosome-Based Delivery Systems

Exosomes derived from mesenchymal stem cells or engineered cells are being explored as natural delivery vehicles for therapeutic genes, offering advantages in terms of biocompatibility and targeting ability.

Advantages for oral applications: Exosomes present several unique advantages for gene delivery in the oral environment, including natural ability to cross biological barriers, reduced immunogenicity compared to viral vectors, natural targeting capabilities, and ability to protect cargo from degradation [34].

### 7.2. Implementation Challenges

#### 7.2.1. Oral-Specific Delivery Barriers

The oral cavity presents a uniquely challenging environment for gene delivery, requiring specialized approaches to overcome multiple barriers [37].

Salivary Barriers:Nuclease activity: Saliva contains DNases and RNases that rapidly degrade nucleic acids.Dilution effects: Continuous salivary flow (0.5–1.5 mL/min) dilutes locally administered vectors.pH variability: Fluctuations between 5.5 and 7.5 affect vector stability and cellular uptake.Solution strategies: Protective formulations, mucoadhesive carriers, and enzyme inhibitors.

Immune Considerations:Mucosal immunity: Active immune surveillance in oral tissues.Previous viral exposure: Pre-existing immunity to common viral vectors.Inflammatory responses: Risk of exacerbating existing oral inflammatory conditions.Mitigation strategies: Immunosuppressive co-delivery, novel vector serotypes, and immunomodulatory approaches.

#### 7.2.2. Manufacturing and Scalability Challenges

The transition from research-grade to clinical-grade gene therapy products presents substantial hurdles. Current manufacturing costs for viral vectors range from USD 100,000 to 500,000 per patient dose, creating accessibility barriers for widespread implementation [38]. Scalable manufacturing platforms, particularly for AAV vectors used in salivary gland applications, require substantial investment in specialized facilities and quality control systems.

#### 7.2.3. Economic Considerations and Healthcare Integration

Cost-effectiveness analyses for oral gene therapy applications are largely absent from the current literature, yet these will be critical for healthcare adoption. The high upfront costs must be justified by long-term clinical benefits, reduced need for conventional therapies, and improved quality of life [131].

#### 7.2.4. Regulatory Pathways

Current regulatory frameworks, while ensuring safety, may require adaptation for oral-specific applications. The confined nature of oral delivery and local action of many oral gene therapies may warrant streamlined approval pathways similar to those being developed for other localized gene therapy applications [132].

Despite the promising advances in gene therapy for oral diseases, several challenges remain:Vector optimization: Developing vectors with improved targeting specificity, reduced immunogenicity, and enhanced transduction efficiency for oral tissues remains a significant challenge [23].Delivery barriers: Overcoming biological barriers to gene delivery in the oral environment, including the mucosal barrier, salivary flow, and microbial biofilms, requires innovative approaches [36].Safety concerns: Addressing safety issues, such as off-target effects, insertional mutagenesis, and immune responses, remains a critical consideration for clinical translation [57].Regulatory hurdles: Navigating the complex regulatory landscape for gene therapy products presents challenges for clinical development and commercialization [133].Cost and accessibility: Ensuring the affordability and accessibility of gene therapy approaches, particularly in resource-limited settings, remains a significant challenge [132].

#### 7.2.5. Patient-Reported Outcomes and Quality of Life Considerations

The success of oral gene therapy must ultimately be measured by meaningful improvements in patients’ daily functioning and quality of life. Current clinical trials in oral gene therapy have primarily focused on biological endpoints, but future studies should incorporate validated patient-reported outcome measures (PROMs) to capture the full therapeutic benefit.

Xerostomia-Specific QOL Measures [131]:Xerostomia Inventory (XI) for subjective dry mouth assessment;Oral Health Impact Profile (OHIP) for functional limitations;Summated Xerostomia Inventory (SXI) for symptom severity;Patient-reported eating, drinking, and speech difficulties.

Cancer Pain QOL Assessments [132]:Brief Pain Inventory (BPI) for pain interference with daily activities;Oral Health-Related Quality of Life (OHRQoL) measures;Functional Assessment of Cancer Therapy—Head and Neck (FACT-H&N);Sleep quality and mood assessments related to pain management.

Periodontal Disease Functional Outcomes [133]:Oral Health Impact Profile-14 (OHIP-14)Patient-reported masticatory function improvements;Aesthetic satisfaction scores;Social confidence and professional impact measures.

Future Trial Design Recommendations:

Current oral gene therapy trials demonstrate a critical need for standardized QOL assessment protocols. Future studies should consider the following:Include baseline and longitudinal QOL measurements;Use validated instruments specific to oral conditions;Assess both disease-specific and general health-related quality of life;Incorporate the patient’s global impression of change scales;Evaluate the durability of QOL improvements beyond biological endpoints.

The integration of robust QOL endpoints will be essential for demonstrating the clinical value of oral gene therapy to patients, healthcare providers, and regulatory agencies.

### 7.3. Ethical Considerations

The advancement of gene therapy for oral diseases also raises important ethical considerations:Informed consent: Ensuring that patients fully understand the risks, benefits, and limitations of gene therapy interventions is essential for ethical clinical practice [134].Equity and access: Addressing disparities in access to gene therapy treatments, particularly for marginalized and underserved populations, is a critical ethical consideration [135].Germline modifications: Distinguishing between somatic and germline genetic modifications and establishing appropriate boundaries for clinical applications remains an important ethical discussion [136].Long-term monitoring: Establishing protocols for long-term monitoring of patients receiving gene therapy treatments to assess delayed effects and outcomes is essential [137].

Table 4 summarizes the current status of clinical trials for gene therapy approaches in oral diseases, highlighting both the progress made and challenges that remain to be addressed.

## 8. Conclusions

Gene therapy represents a revolutionary medical approach that targets disease mechanisms fundamentally differently from symptomatic treatments. It holds promise for the treatment of genetic disorders and has potential applications in cancer, pain management, dental caries, periodontal disease, and xerostomia. However, current gene transfer technologies require further refinement before full clinical implementation.

### 8.1. Current State Assessment

The field of gene therapy for oral diseases has witnessed significant progress over the past decade, with advances in vector design, delivery methods, and therapeutic strategies. Clinical trials, particularly in the areas of salivary gland gene therapy for xerostomia and cancer pain management, have demonstrated safety and promising efficacy outcomes. The development of novel gene delivery systems, combined with advances in biomaterials and regenerative approaches, offers exciting possibilities for treating periodontal diseases and promoting oral tissue regeneration.

### 8.2. Translational Outlook

Manufacturing and scalability challenges: The transition from research-grade to clinical-grade gene therapy products presents substantial hurdles. Current manufacturing costs for viral vectors range from USD 100,000 to 500,000 per patient dose, creating accessibility barriers for widespread implementation. Scalable manufacturing platforms, particularly for AAV vectors used in salivary gland applications, require substantial investment in specialized facilities and quality control systems.

Economic considerations and healthcare integration: Cost-effectiveness analyses for oral gene therapy applications are largely absent from the current literature, yet these will be critical for healthcare adoption. The high upfront costs must be justified by long-term clinical benefits, reduced need for conventional therapies, and improved quality of life. For chronic conditions like xerostomia affecting cancer survivors, the economic argument may be compelling given the lifelong nature of conventional management.

Regulatory pathway optimization: Current regulatory frameworks, while ensuring safety, may require adaptation for oral-specific applications. The confined nature of oral delivery and local action of many oral gene therapies may warrant streamlined approval pathways similar to those being developed for other localized gene therapy applications.

### 8.3. Future Integration with Precision Medicine

The convergence of gene therapy with precision medicine approaches offers particular promise for oral applications. Genetic profiling of periodontal disease susceptibility, cancer pain sensitivity, and salivary gland dysfunction may enable personalized gene therapy approaches that optimize efficacy while minimizing adverse effects.

The integration of gene therapy with other emerging technologies, including 3D bioprinting for delivery scaffold production, artificial intelligence for treatment optimization, and nanotechnology for enhanced targeting, represents the next frontier in oral gene therapy development.

Looking ahead, the integration of gene therapy with other emerging technologies, such as CRISPR-Cas9 gene editing, stem cell-based approaches, and precision medicine strategies, holds promise for addressing the complex challenges of oral diseases. While significant hurdles remain, including vector optimization, delivery barriers, safety concerns, and regulatory considerations, the continued advancement of gene therapy technologies offers hope for more effective, targeted, and personalized treatments for oral diseases in the future.

Despite these challenges, the unique advantages of oral gene therapy—including accessible delivery sites, confined treatment areas, and the potential for both local and systemic benefits—position this field for significant clinical impact. Continued collaboration among researchers, clinicians, industry partners, and regulatory agencies will be essential for realizing the full therapeutic potential of gene therapy in oral healthcare.

Advancing understanding of disease mechanisms and improving gene transfer techniques are crucial for establishing safe and effective treatments across both dental and broader medical fields. Collaborative efforts among researchers, clinicians, regulatory agencies, and industry partners will be essential for translating promising preclinical findings into clinical applications that can benefit patients with oral diseases.

## Figures and Tables

**Figure 1 pharmaceutics-17-00859-f001:**
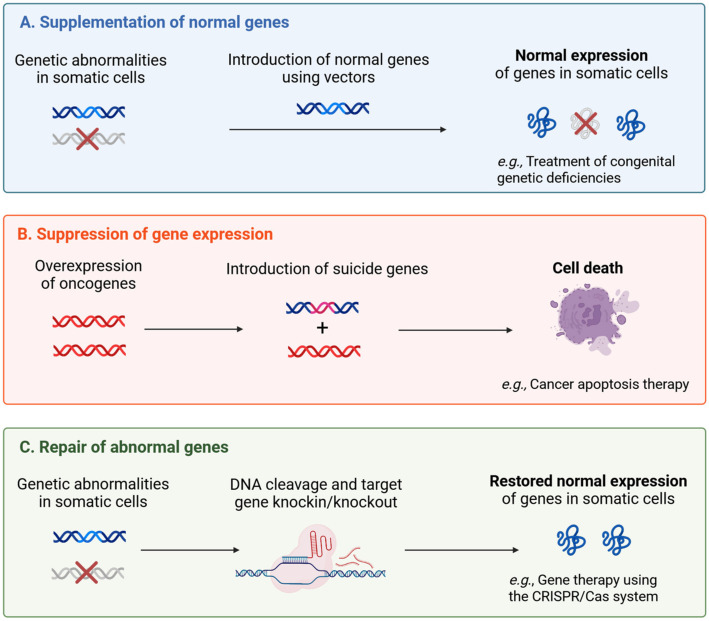
Three basic concepts of gene therapy. (**A**) Supplementing normal genes for genetic disorders. (**B**) Suppressing overexpressed cancer genes using apoptosis or suppressor genes. (**C**) Fundamentally controlling gene expression through genome editing, like clustered regularly interspaced short palindromic repeats (CRISPR) and CRISPR-associated 9 (CRISPR-Cas9).

**Figure 2 pharmaceutics-17-00859-f002:**
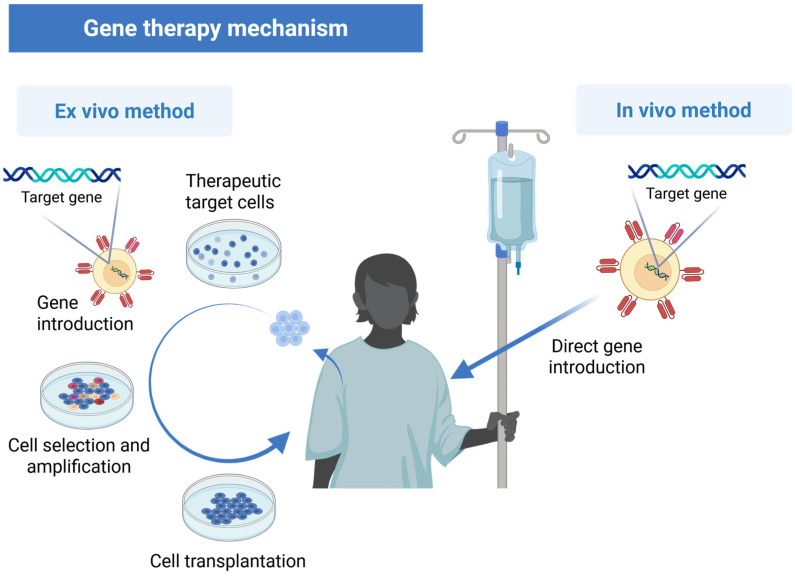
Gene transfer has two methods: ex vivo—cells extracted and manipulated outside the patient’s body; in vivo—direct vector injection for gene transfer within the body.

**Table 1 pharmaceutics-17-00859-t001:** Biological properties of commonly used viral vectors for gene therapy.

Vector Type	Genome	Packaging Capacity	Integration	Duration of Expression	Main Advantages	Main Limitations	Oral Applications	Key References
Adenovirus	dsDNA	8–36 kb	Non-integrating	Transient (days to weeks)	High transduction efficiency; Transduces dividing and non-dividing cells; Easy to produce in high titers	High immunogenicity; Pre-existing immunity common	Oral cancer; Xerostomia; Periodontal diseases	[12,13,14,15]
Adeno-associated virus (AAV)	ssDNA	4.7 kb	Rarely integrating	Long-term (months to years)	Low immunogenicity; Transduces dividing and non-dividing cells; Multiple serotypes for tissue targeting	Limited packaging capacity; Slow onset of expression	Salivary gland disorders; Periodontal regeneration; Cancer pain	[16,17,18,19]
Retrovirus	ssRNA	8 kb	Integrating	Long-term (years)	Stable gene expression due to integration	Only transduces dividing cells; Risk of insertional mutagenesis	Primarily ex vivo approaches for oral cancer	[20,21]
Lentivirus	ssRNA	8 kb	Integrating	Long-term (years)	Transduces dividing and non-dividing cells; Lower immunogenicity than adenovirus	Risk of insertional mutagenesis; Complex production	Ex vivo modification of cells for periodontal regeneration	[22,23]
Herpes simplex virus	dsDNA	>30 kb	Non-integrating	Transient	Large packaging capacity; Natural neurotropism	Cytotoxicity; Complex genome	Oral cancer pain; Oncolytic therapy for oral cancer	[24]
Vaccinia virus	dsDNA	>25 kb	Non-integrating	Transient	Large packaging capacity; Strong immunogenicity	Limited targeting specificity; Safety concerns	Cancer immunotherapy	[25]

**Table 2 pharmaceutics-17-00859-t002:** Biological properties of commonly used non-viral vectors for gene therapy.

Vector Type	Composition	Packaging Capacity	Transfection Efficiency	Duration of Expression	Main Advantages	Main Limitations	Oral Applications	Key References
Lipid-based systems	Cationic lipids, helper lipids, PEG-lipids	Unlimited	Moderate	Short-term (days)	Low immunogenicity; Easy to produce; Suitable for various nucleic acids	Serum instability; Cytotoxicity at high concentrations	Oral cancer; Mucosal delivery	[28,29,30]
Polymer-based systems	PEI, PLL, PAMAM, chitosan	Unlimited	Low to moderate	Short-term (days)	Low cost; Versatility; Stability	Lower efficiency than viral vectors; Potential cytotoxicity	Periodontal diseases; Controlled release applications	[26,28,31]
Lipopoly-plex	Combination of lipids and polymers	Unlimited	Moderate to high	Short-term (days)	Improved efficiency compared to individual components; Reduced toxicity	Complex formulation; Batch-to-batch variability	Cancer pain; Periodontal regeneration	[28,29]
Cell-penetrating peptides (CPP)	Short peptides with membrane-penetrating properties	Limited by complexation	Moderate	Short-term (days)	Enhanced cellular uptake; Low immunogenicity	Limited stability; High cost	Targeting oral cancer cells; Salivary gland delivery	[28,32]
Inorganic nanoparticles	Gold, silica, calcium phosphate	Variable	Low to moderate	Short-term (days)	Stability; Surface functionalization options	Potential toxicity; Variable efficiency	Dental hard tissue regeneration; Antimicrobial applications	[33]
Exosomes	Natural membrane vesicles	Limited by vesicle size	Variable	Short to medium term	Low immunogenicity; Natural targeting	Complex isolation; Standardization challenges	Emerging applications in oral inflammatory diseases	[34]

**Table 3 pharmaceutics-17-00859-t003:** Vector selection for specific oral applications with references.

Oral Application	Preferred Vector Types	Key Considerations	Key References
Oral Cancer	Adenoviral, Oncolytic viral vectors	High transduction efficiency, tumor selectivity	[39,40,41,42]
Cancer Pain	AAV, Non-viral	Neurotropism, localized delivery, safety	[43,44,45]
Xerostomia	AAV, Adenoviral	Duration of expression, immunogenicity	[14,15,18,19]
Periodontal Diseases	Non-viral, AAV with biomaterials	Scaffold integration, spatial control	[46,47,48,49]
Dental Caries	Non-viral, Bacteriophage	Mucosal delivery, microbiome targeting	[50,51]

**Table 4 pharmaceutics-17-00859-t004:** Current clinical trials of gene therapy for oral diseases.

Disease	Vector	Therapeutic Gene	Phase	Trial ID	Status	Primary Outcomes	Key Findings	Key References
Radiation-induced xerostomia	Adenovirus	Aquaporin-1 (AQP1)	I/II	NCT02446249	Completed	Safety; Change in parotid salivary flow rate	Increased parotid flow rate in 5/11 patients; No serious adverse events	[14,105]
Radiation-induced xerostomia	AAV2	Aquaporin-1 (AQP1)	I	NCT02749448	Active	Safety; Vector shedding; Preliminary efficacy	Ongoing; Preliminary results show improved safety profile compared to adenoviral vectors	[18,19]
Head and neck squamous cell carcinoma	Oncolytic adenovirus (ONCOS-102)	GM-CSF	I	NCT01598129	Completed	Maximum tolerated dose; Safety	Acceptable safety profile; Evidence of immune activation in tumor microenvironment	[76,77,78]
Head and neck squamous cell carcinoma	Oncolytic herpes simplex virus (HSV-1716)	-	I	NCT00931931	Completed	Safety; Virus replication in tumor	Well-tolerated; Evidence of selective replication in tumor tissues	[24,25]
Oral cancer pain	Non-viral CPP/lipid	μ-opioid receptor	Pre-clinical	-	-	Safety; Change in pain scores	Preliminary data indicates significant reduction in pain scores; No serious adverse events	[43,44]
Head and neck cancer	mRNA lipid nanoparticles	HPV E6/E7 antigens	I	NCT03418480	Active	Safety; Immunogenicity	Ongoing; Early results show induction of antigen-specific T cell responses	[69,85]
Sjögren’s syndrome	AAV	VIP	Pre-clinical	-	-	Reduction in inflammatory markers; Restoration of salivary flow	60% improvement in salivary flow in mouse models; Reduction in lymphocytic infiltration	[108]
Periodontal diseases	Plasmid/polymer	PDGF-B	I/II	NCT00540423	Completed	Safety; Clinical attachment level gain	Safe with no significant adverse events; 1.3–2.3 mm mean clinical attachment level gain	[46,47]
Advanced head and neck cancer	Retrovirus	Wild-type p53	II	NCT00004224	Completed	Tumor response rate	26% objective response rate; Median survival of 7.5 months vs. 4.5 months in control group	[66,67]
Recurrent head and neck cancer	Adenovirus (INGN 201)	p53	II/III	NCT00041626	Completed	Locoregional tumor control	40% local tumor control rate when combined with chemotherapy vs. 20% with chemotherapy alone	[66,67,82]
Dental caries	Plasmid DNA/cationic liposome	Antimicrobial peptide LL-37	Pre-clinical	-	-	Reduction in S. mutans colonization	85% reduction in *S. mutans* biofilm formation in ex vivo models	[118]
Peri-implantitis	Adenovirus	BMP-7	Pre-clinical	-	-	Bone regeneration around implants	40% greater bone fill in animal models compared to control treatments	[48]

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
