# Peer review of "Application of Gene Therapy to Oral Diseases"

_pharmaceutics, 2025, doi:10.3390/pharmaceutics17070859_

Round 1

Reviewer 1 Report

Comments and Suggestions for Authors

The manuscript provides a comprehensive overview of gene therapy applications in oral diseases, addressing key areas with current research and clinical trials. Despite the content is valuable, several improvements are needed to enhance clarity, rigor, and scholarly impact.

1.Section Transitions: Strengthen links between sections (e.g., add a brief summary after the Introduction to outline the review’s structure).

2.Subsection 3.4 (mRNA Therapeutics): Merge redundant paragraphs on delivery innovations to avoid repetition.

3.Conclusion: Emphasize translational challenges (e.g., scalability, cost) more prominently.

Punctuation & Formatting

4.Table 1: Use consistent hyphenation (e.g., "Non-integrating" vs. "Rarely integrating").

5.Section 5.5.2: Add commas in "pH-responsive, temperature-sensitive, and enzyme-responsive hydrogels" for clarity.

6.References: Ensure consistent formatting (e.g., journal abbreviations, italics for Mol. Ther.).

7.Use "CRISPR-Cas9" uniformly (avoid variations like "CRISPR/Cas"). Standardize "non-viral" vs. "nonviral" throughout the text.

8.Address patient-reported outcomes in clinical trials (e.g., quality-of-life metrics for xerostomia)

Author Response

Response to Reviewer 1

Overall Assessment: We sincerely thank Reviewer 1 for their constructive feedback and positive 3-star assessment. We have carefully addressed all eight points raised, significantly improving manuscript clarity, organization, and scientific rigor.

Point 1: Section Transitions

Reviewer Comment: “Strengthen links between sections (e.g., add a summary after the Introduction to outline the review's structure).”

Response: We have completely addressed this concern by adding a comprehensive new Section 1.3, “Review Structure and Roadmap,” that clearly outlines how each section builds upon previous content. This roadmap explains the systematic progression from fundamental gene therapy principles (Section 2) through specific disease applications (Sections 3-6) to future directions and challenges (Section 7).

Specific Changes Made:

  • Added Section 1.3 (lines 47-58): Detailed roadmap explaining the review's structure
  • Enhanced transitions: Each major section now begins with connecting language
  • Improved flow: Clear progression from basic concepts to clinical applications
  • Reader guidance: Explicit explanation of how sections interconnect

Point 2: Subsection 3.4 (mRNA Therapeutics): Merge redundant paragraphs

Reviewer Comment: “Merge redundant paragraphs on delivery innovations to avoid repetition.”

Response: We have streamlined Section 3.4 by consolidating overlapping content in the delivery innovations discussion. The revised Section 3.4.2 now presents a concise, integrated discussion of delivery systems without redundancy.

Specific Changes Made:

  • Merged content in Section 3.4.2: Combined redundant delivery discussions into a single coherent paragraph
  • Eliminated repetition: Removed overlapping information while maintaining comprehensive coverage
  • Improved clarity: Streamlined presentation focuses on the distinct advantages of each approach
  • Enhanced efficiency: More concise yet thorough coverage of mRNA delivery innovations

Point 3: Conclusion: Emphasize translational challenges

Reviewer Comment: “Emphasize translational challenges (e.g., scalability, cost) more prominently.”

Response: We have significantly expanded the conclusion with a new dedicated “Section 8.2 Translational Outlook” that comprehensively addresses practical implementation challenges as requested.

Specific Changes Made:

  • Added Section 8.2: Detailed discussion of translational challenges, including:
    • Manufacturing and scalability challenges (cost analysis: $100,000-500,000 per dose)
    • Economic considerations and healthcare integration requirements
    • Regulatory pathway optimization for oral-specific applications
  • Enhanced emphasis: Translational barriers are now prominently featured in the conclusion
  • Practical focus: Real-world implementation challenges clearly articulated
  • Future perspective: Integration with precision medicine and emerging technologies

Point 4: Table 1: Consistent hyphenation

Reviewer Comment: “Use consistent hyphenation (e.g., 'Non-integrating' vs. 'Rarely integrating').”

Response: We have standardized terminology throughout Table 1 for consistency and clarity.

Specific Changes Made:

  • Standardized terminology: “Non-integrating” used for adenovirus, herpes simplex virus, and vaccinia virus
  • Precise terminology: “Rarely integrating (it’s not often)” specifically used for AAV vectors
  • Consistent application: Uniform hyphenation applied throughout all tables
  • Scientific accuracy: Terminology reflects actual biological properties

Point 5: Section 5.5.2: Punctuation clarity

Reviewer Comment: “Add commas in ‘pH-responsive, temperature-sensitive, and enzyme-responsive hydrogels’ for clarity.”

Response: We have corrected punctuation throughout the manuscript for improved readability.

Specific Changes Made:

  • Corrected punctuation: “pH-responsive, temperature-sensitive, and enzyme-responsive hydrogels”
  • Enhanced clarity: Proper comma usage implemented throughout
  • Consistency check: Similar punctuation issues resolved manuscript-wide
  • Professional presentation: Improved grammatical accuracy

Point 6: References: Consistent formatting

Reviewer Comment: “Ensure consistent formatting (e.g., journal abbreviations, italics for Mol. Ther.).”

Response: We have implemented comprehensive reference formatting standardization across all 178 citations.

Specific Changes Made:

  • Uniform journal abbreviations: Consistent use of standard abbreviations (e.g., “Mol. Ther.”)
  • Proper italicization: All journal names are properly italicized
  • DOI formatting: Consistent DOI presentation where available
  • Author formatting: Standardized author name presentation
  • Volume/page consistency: Uniform formatting for all bibliographic elements

Point 7: CRISPR terminology standardization

Reviewer Comment: “Use ‘CRISPR-Cas9’ uniformly (avoid variations like ‘CRISPR/Cas’).”

Response: We have standardized CRISPR terminology throughout the manuscript for consistency.

Specific Changes Made:

  • Uniform terminology: “CRISPR-Cas9” used consistently throughout (e.g., Section 2.1, 3.2, 7.1.1)
  • Scientific accuracy: Specific variants used only when appropriate (e.g., “Cas12a” when specifically referenced)
  • Definition clarity: Full expansion provided at first use: “clustered regularly interspaced short palindromic repeats (CRISPR)/CRISPR-associated 9 (CRISPR-Cas9)”
  • Professional presentation: Consistent scientific nomenclature

Point 8: Clinical trials outcomes

Reviewer Comment: “Address patient-reported outcomes in clinical trials (e.g., quality-of-life metrics for xerostomia).”

Response: We have enhanced our clinical trials discussion to include patient-reported outcomes and quality-of-life considerations as requested.

Specific Changes Made:

  • Enhanced Table 4: Now includes “Key Findings” column that emphasizes patient-relevant outcomes such as “Increased parotid flow rate in 5/11 patients” for xerostomia trials and “significant reduction in pain scores” for cancer pain studies, focusing on clinically meaningful endpoints rather than purely biological markers
  • Added comprehensive QOL discussion: New content added to “Section 7.2.5 Patient-Reported Outcomes and Quality of Life Considerations,” specifically addressing patient-reported outcomes integration, stating: “Future oral gene therapy trials must prioritize patient-reported quality-of-life measures alongside biological endpoints. Validated instruments such as the Xerostomia Inventory (XI), Brief Pain Inventory (BPI), and Oral Health Impact Profile (OHIP) should be systematically incorporated to capture meaningful improvements in daily functioning, comfort, and social well-being.”
  • Clinical trial design recommendations: The new content explicitly addresses the knowledge gap, noting that “Current trials have demonstrated biological efficacy but lack comprehensive assessment of patient-perceived benefits, representing a critical knowledge gap for clinical translation.”
  • Patient-centered focus: Throughout the manuscript, we emphasize functional outcomes that matter to patients - speech, eating, swallowing for xerostomia; pain interference with daily activities for cancer pain; and masticatory function for periodontal applications
  • Future trial guidance: We now explicitly state, "This patient-centered approach will be essential for demonstrating clinical value and supporting regulatory approval of oral gene therapies.”

Reviewer 2 Report

Comments and Suggestions for Authors

Seiichi Yamano et al reviewed current applications and future prospects of gene therapy in dentistry, and mainly focused on five key areas: oral cancer, cancer-related pain, xerostomia (dry mouth), dental caries, and periodontal disease. This review addresses an important and timely topic, exploring the applications of gene therapy in oral diseases. However, I think this manuscript needs major revision and is not suitable for publishing in Pharmaceuticals in its current form. Here are some comments of this manuscript for authors.

  1. A more detailed discussion on rationale for vector selection (e.g., AAV vs. lentiviral vectors in oral tissue tropism) would strengthen the technical foundation.
  2. The barriers to clinical translation (e.g., oral mucosal immunity, vector stability in saliva, off-target effects) warrant further elaboration to provide a balanced perspective.
  3. A dedicated section on oral microenvironment-specific challenges (e.g., salivary nucleases, pH) could highlight unique considerations for gene delivery in this field.
  4. While Phase I trials are mentioned, their limitations (small cohorts, short-term follow-up) should be explicitly addressed.
  5. Potential mechanisms of treatment resistance in oral cancer gene therapy could be briefly discussed to balance efficacy claims.
  6. The authors should propose some oral-specific delivery systems (e.g., stimuli-responsive hydrogels, targeted nanoparticles) as a priority for future research.
  7.  The structure of the article is a bit chaotic, there are numbers under the second (such as 3.1, 3.2, 3.3, 4.1, 4.2, 4.3, 5.1, 5.2, 5.3, 5.4) and third (3.4.1, 3.4.2, 5.5.1 and 5.5.2) level headings, and should be revised.

Author Response

Response to Reviewer 2

Overall Assessment: We deeply appreciate Reviewer 2’s detailed technical feedback and strong 4-star rating. We have implemented comprehensive improvements addressing all seven major points, significantly enhancing the manuscript's technical depth and scientific rigor.

Point 1: Vector Selection Rationale

Reviewer Comment: “A more detailed discussion on rationale for vector selection (e.g., AAV vs. lentiviral vectors in oral tissue tropism) would strengthen the technical foundation.”

Response: We have completely transformed Section 2.3.2 into a comprehensive vector selection framework that provides detailed technical rationale for oral applications.

Specific Changes Made:

  • Enhanced Section 2.3.2: Now includes comprehensive “Decision Framework for Vector Selection”
  • Primary considerations: Five key factors explicitly listed:
    1. Target tissue characteristics (epithelial vs. mesenchymal vs. neural)
    2. Expression requirements (transient vs. sustained)
    3. Accessibility (surface vs. deep tissue)
    4. Safety profile (local vs. systemic exposure)
    5. Manufacturing feasibility (clinical-grade production)
  • Tissue-specific rationale: Detailed explanations for oral cancer, salivary gland, and periodontal applications
  • Technical depth: AAV vs. adenoviral considerations specific to the oral environment
  • Enhanced Table 3: Now includes specific delivery routes and detailed key considerations

Point 2: Clinical Translation Barriers

Reviewer Comment: “The barriers to clinical translation (e.g., oral mucosal immunity, vector stability in saliva, off-target effects) warrant further elaboration.”

Response: We have added two new dedicated sections that comprehensively address oral-specific translation barriers.

Specific Changes Made:

  • New Section 2.3.3: “Oral-Specific Delivery Considerations” with detailed discussion of:
    • Salivary barriers (DNases/RNases, pH variability, dilution effects)
    • Mucosal barrier penetration challenges
    • Enhancement approaches and solution strategies
  • New Section 7.2.1: “Oral-Specific Delivery Barriers” with expanded coverage of:
    • Immune considerations (mucosal immunity, pre-existing immunity)
    • Biofilm challenges and bacterial interference
    • Mitigation strategies for each barrier type
  • Technical solutions: Specific approaches to overcome each identified barrier
  • Clinical relevance: Direct application to current and future oral gene therapy protocols

Point 3: Microenvironment-Specific Challenges

Reviewer Comment: “A dedicated section on microenvironment-specific challenges (e.g., salivary nucleases, pH) could highlight unique considerations.”

Response: We have created comprehensive content specifically addressing the unique oral microenvironment challenges.

Specific Changes Made:

  • Detailed oral environment analysis: pH variability (5.5-7.5), salivary flow rates (0.5-1.5 mL/min), nuclease activity
  • Unique challenge identification: Factors specific to the oral cavity vs. other delivery sites
  • Environmental considerations: Temperature, mechanical forces, bacterial interactions
  • Practical implications: How do these factors affect vector design and delivery protocols
  • Solution strategies: Protective formulations, mucoadhesive carriers, enzyme inhibitors

Point 4: Phase I Trial Limitations

Reviewer Comment: “While Phase I trials are mentioned, their limitations (small cohorts, short-term follow-up) should be explicitly addressed.”

Response: We have added a comprehensive critical analysis of current clinical trial limitations following enhanced Table 4.

Specific Changes Made:

  • Enhanced Table 4: Now includes detailed analysis of trial limitations for each study
  • Critical analysis section: Systematic evaluation of:
    • Sample size limitations (most trials ≤15 patients)
    • Follow-up inadequacies (median <6 months)
    • Endpoint limitations (reliance on surrogate endpoints)
    • Design considerations for future trials
  • Specific recommendations: Minimum 50-100 patients for Phase II, 2-5 year follow-up, standardized outcome measures
  • Clinical context: How limitations affect the interpretation of results and clinical translation

Point 5: Treatment Resistance Mechanisms

Reviewer Comment: “Potential mechanisms of treatment resistance in oral cancer gene therapy could be briefly discussed to balance efficacy claims.”

Response: We have added a comprehensive new Section 3.5, “Resistance Mechanisms and Strategies to Overcome Them,” that thoroughly addresses this critical topic.

Specific Changes Made:

  • Major new section (3.5): Comprehensive coverage of resistance mechanisms, including:
    • Vector-related resistance (immune clearance, degradation)
    • Cellular resistance (reduced uptake, enhanced DNA repair)
    • Microenvironment resistance (immunosuppressive environment, hypoxia)
    • Genetic resistance (tumor heterogeneity, epigenetic silencing)
  • Solution strategies: Specific approaches to overcome each resistance type
  • Clinical relevance: How resistance affects treatment planning and combination approaches
  • Evidence-based: Well-referenced with current literature (References 96-99)

Point 6: Oral-Specific Delivery Systems

Reviewer Comment: “The authors should propose some oral-specific delivery systems (e.g., stimuli-responsive hydrogels, targeted nanoparticles) as a priority for future research.”

Response: We have significantly expanded the discussion of innovative oral-specific delivery systems throughout multiple sections.

Specific Changes Made:

  • Enhanced Section 6.3: Detailed discussion of advanced biomaterials, including:
    • Smart hydrogels with stimuli-responsive properties
    • 3D-printed scaffolds for patient-specific applications
    • Nanoparticle-based delivery systems
  • New content in Section 7.1.3: Exosome-based delivery systems with oral-specific advantages
  • Section 5.5: Advanced lubricant formulations and smart hydrogel systems
  • Future directions: Specific recommendations for oral-optimized delivery platforms
  • Technical specifications: Detailed mechanisms and advantages for oral applications

Point 7: Article Structure

Reviewer Comment: “The structure of the article is a bit chaotic, there are numbers under the second (such as 3.1, 3.2, 3.3, 4.1, 4.2, 4.3, 5.1, 5.2, 5.3, 5.4) and third (3.4.1, 3.4.2, 5.5.1 and 5.5.2) level headings, and should be revised.”

Response: We have completely reorganized the manuscript with a professional, consistent hierarchical numbering system throughout.

Specific Changes Made:

  • Consistent 8-section structure: Clear progression from 1-8 with logical organization
  • Uniform subsection numbering: Consistent X.Y.Z format throughout (e.g., 1.1, 1.2, 2.1, 2.2.1, 2.2.2, 2.2.3)
  • Logical hierarchy: Proper nesting of subsections with appropriate numbering depth
  • Professional presentation: Manuscript now follows standard academic formatting conventions
  • Enhanced navigation: Clear section structure enables easy reference and comprehension
  • Eliminated chaos: All numbering inconsistencies resolved with systematic organization

Reviewer 3 Report

Comments and Suggestions for Authors

Reviewer Comments on the Manuscript: “Application of Gene Therapy to Oral Diseases”

The manuscript provides a well-structured overview of gene therapy, linking its relevance to both general medicine and dentistry. It effectively sets the groundwork for further exploration into therapeutic innovations and personalized medicine in future research. However, to enhance the manuscript's readability, accuracy, and academic rigor, I recommend the following revisions:

  1. Missing References in the Introduction (Lines 32–35):
    The discussion of gene therapy applications in adenosine deaminase deficiency, muscular atrophy, hemophilia, and Parkinson’s disease requires proper citations. Please include relevant peer-reviewed references to support these examples.
  2. Lack of References in Paragraph II of the Introduction (Lines 36–37):
    This section also lacks citations to substantiate the claims made. Adding appropriate references will strengthen the credibility of the manuscript.
  3. “What is Gene Therapy” Section (Lines 43–52):
    This section should be made more concise and focused. Consider elaborating briefly on the mechanism of gene therapy—such as gene delivery techniques, target cell types, and gene expression regulation. Additionally, the content in lines 43–52, which includes background information on gene mutations and their implications in diseases, would be more appropriately placed in the introduction to improve logical flow.
  4. Gene Transfer Methods:
    The discussion of gene transfer techniques (biological, chemical, and physical methods) should be supported with up-to-date and authoritative references. Please ensure that each method includes citations to relevant studies or reviews.
  5. Tables 1, 2, and 3 – Missing Reference Information:
    Each table should include an additional column citing the sources of the data presented. Without references, it is difficult to verify the validity and origin of the information provided. This is essential for maintaining transparency and academic integrity.

Author Response

Response to Reviewer 3

Overall Assessment: We thank Reviewer 3 for their critical 2-star assessment that highlighted fundamental issues requiring attention. We have comprehensively addressed all five major concerns, transforming the manuscript into a well-referenced, organized, and transparent scientific review.

Point 1: Missing References in Introduction (Lines 32-35)

Reviewer Comment: “The discussion of gene therapy applications in adenosine deaminase deficiency, muscular atrophy, hemophilia, and Parkinson’s disease requires proper citations. Please include relevant peer-reviewed references to support these examples.”

Response: We have added comprehensive, high-quality citations for all mentioned therapeutic applications in the Introduction.

Specific Changes Made:

  • Reference 1: Blaese et al. (1995) Science - for adenosine deaminase deficiency (first human gene therapy trial)
  • Reference 3: Day et al. (2021) Lancet Neurol. - for spinal muscular atrophy (Zolgensma)
  • Reference 4: George et al. (2020) Mol. Ther. - for hemophilia gene therapy
  • Reference 5: Muramatsu et al. (2010) Mol. Ther. - for Parkinson’s disease, AAV therapy
  • High-impact sources: All references from top-tier journals (Science, Lancet, Molecular Therapy)
  • Current relevance: Recent publications demonstrating the current state of the field

Point 2: Lack of References in Paragraph II of Introduction (Lines 36-37)

Reviewer Comment: “This section also lacks citations to substantiate the claims made. Adding appropriate references will strengthen the credibility of the manuscript.”

Response: We have added authoritative references supporting epidemiological claims and gene therapy rationale for oral diseases.

Specific Changes Made:

  • Reference 6: Peres et al. (2019), Lancet - authoritative source for global oral disease burden (3.5 billion people affected)
  • Reference 7: Jin et al. (2022) Front. Genet. - A recent comprehensive review of gene therapy for periodontal tissue engineering
  • Evidence-based claims: All epidemiological data are now properly supported
  • Credibility enhancement: High-impact journal sources strengthen manuscript authority

Point 3: “What is Gene Therapy?” Section (Lines 43-52)

Reviewer Comment: “This section should be made more concise and focused. Consider elaborating briefly on the mechanism of gene therapy, such as gene delivery techniques, target cell types, and gene expression regulation. Additionally, the content in lines 43-52, which includes background information on gene mutations and their implications in diseases, would be more appropriately placed in the introduction to improve logical flow.”

Response: We have completely restructured Section 2.1 to be more concise and technically focused while moving the general background to the Introduction.

Specific Changes Made:

  • Streamlined Section 2.1: Now focuses specifically on gene therapy principles and mechanisms
  • Enhanced technical content: Added detailed discussion of three primary mechanisms:
    1. Supplementing missing/defective genes
    2. Suppressing overexpressed disease-causing genes
    3. Controlling gene expression through genome editing (CRISPR-Cas9)
  • Improved organization: General disease background relocated to Introduction Section 1.1
  • Logical flow: Better progression from basic concepts to technical applications
  • Professional focus: Section now appropriate for technical audience

Point 4: Gene Transfer Methods

Reviewer Comment: “The discussion of gene transfer techniques (biological, chemical, and physical methods) should be supported with up-to-date and authoritative references. Please ensure that each method includes citations to relevant studies or reviews.”

Response: We have added comprehensive, authoritative citations for all gene transfer technologies with current, high-quality references.

Specific Changes Made:

  • Comprehensive referencing: References 10-33 covering all gene transfer methodologies
  • Viral vectors: References 12-23, including recent advances in AAV, lentiviral, and adenoviral systems
  • Non-viral vectors: References 24-32 covering lipid nanoparticles, polymeric systems, and hybrid approaches
  • Physical methods: Reference 33 for electroporation and other direct delivery techniques
  • Current literature: All references from 2010-2024, emphasizing recent advances
  • Authoritative sources: High-impact journals and comprehensive reviews

Point 5: Tables 1, 2, and 3 - Missing Reference Information

Reviewer Comment: “Each table should include an additional column citing the sources of the data presented. Without references, it is difficult to verify the validity and origin of the information provided. This is essential for maintaining transparency and academic integrity.”

Response: We have completely revised all tables to include comprehensive reference columns, ensuring full transparency and verifiability of all presented data.

Specific Changes Made:

  • Table 1 (Viral Vectors): Added "Key References" column with 3-4 citations per vector type
    • Adenovirus: [12,13,14,15]
    • AAV: [13,16,17,18]
    • Retrovirus: [19,20]
    • Lentivirus: [19,21]
    • HSV: [22,23]
    • Vaccinia: [22]
  • Table 2 (Non-viral Vectors): Added reference column with 2-4 citations per system
    • Lipid-based: [25,26,27]
    • Polymer-based: [25,26,28]
    • Lipopoly-plex: [25,26]
    • CPP: [25,29]
    • Inorganic: [30]
    • Exosomes: [31,32]
  • Table 3 (Vector Selection): Added references for each oral application
    • Oral Cancer: [38,39,40,41]
    • Cancer Pain: [42,43,44]
    • Xerostomia: [45,46,47,48]
    • Periodontal: [49,50,51,52]
    • Dental Caries: [53,54]
  • Table 4 (Clinical Trials): Enhanced with comprehensive reference columns
    • Each trial entry includes 3-4 supporting references
    • Complete transparency in data sources
  • Academic integrity: Full verifiability of all tabulated information
  • Professional standard: Tables now meet the highest academic standards

Round 2

Reviewer 1 Report

Comments and Suggestions for Authors

The revision is fine, and can be accepted

Reviewer 2 Report

Comments and Suggestions for Authors

The authors addressed most of my concerns.